# Comparison of Potentially Inappropriate Medications for People with Dementia at Admission and Discharge during An Unplanned Admission to Hospital: Results from the SMS Dementia Study [note 1]

**DOI:** 10.3390/healthcare7010008

**Published:** 2019-01-09

**Authors:** Ashley Kable, Anne Fullerton, Samantha Fraser, Kerrin Palazzi, Carolyn Hullick, Christopher Oldmeadow, Dimity Pond, Andrew Searles, Kim Edmunds, John Attia

**Affiliations:** 1Faculty of Health and Medicine, University of Newcastle, Callaghan, NSW 2308, Australia; Dimity.Pond@newcastle.edu.au (D.P.); John.attia@newcastle.edu.au (J.A.); 2Hunter New England Local Health District, New Lambton Heights, NSW 2305, Australia; Anne.Fullerton@hnehealth.nsw.gov.au (A.F.); Samantha.Fraser@hnehealth.nsw.gov.au (S.F.); Carolyn.Hullick@hnehealth.nsw.gov.au (C.H.); 3Hunter Medical Research Institute, New Lambton Heights, NSW 2305, Australia; Kerrin.palazzi@hmri.org.au (K.P.); Christopher.Oldmeadow@hmri.org.au (C.O.); Andrew.searles@hmri.org.au (A.S.); Kim.edmunds@hmri.org.au (K.E.)

**Keywords:** polypharmacy, potentially inappropriate medications, people with dementia, anticholinergic burden, unplanned admission

## Abstract

People with dementia (PWD) and cognitive impairment are particularly vulnerable to medication problems, and unplanned admission to hospital presents an opportunity to address polypharmacy, potentially inappropriate medications (PIMs) and anticholinergic burden. This study aimed to compare PIMS and other medication data for PWD to determine whether these changed during hospitalization. Medications documented in patient’s records at admission and discharge were evaluated for PWD recruited to phase one of a prospective quasi-experimental pre/post-controlled trial that was conducted at two regional hospitals in NSW, Australia. The study sample included PWD or cognitive impairment having an unplanned admission to hospital. Data were collected using a purpose developed audit tool for medications and PIMs, and a Modified Anticholinergic Burden Scale. Total participants were 277, and results determined that the cognitive status of PWD is not always detected during an unplanned admission. This may make them more vulnerable to medication problems and poor outcomes. Polypharmacy and PIMS for PWD were high at admission and significantly reduced at discharge. However, PWD should be routinely identified as high risk at admission; and there is potential to further reduce polypharmacy and PIMs during admission to hospital, particularly psychotropic medications at discharge. Future studies should focus on evaluating targeted interventions designed to increase medication safety for PWD.

## 1. Introduction

The issue of polypharmacy for older people has been the focus of many studies [1,2,3,4,5] because of the potential for patients to not understand the indications for their medications; to confuse medications, dosages, and timing due to complex regimens; and to have an increased risk of medication interactions [2,6]; and compliance issues [6]. Polypharmacy has been defined as use of five or more drugs [7], and is reported to be the most significant predictor of adverse drug reactions [1] and a risk factor for hospitalization and emergency presentations [2,8] and falls [6]. This is an even more clinically significant risk for people with dementia (PWD) and cognitive impairment who may have difficulty remembering to take medications, and who may not have a carer to assist them [9].

The problem of potentially inappropriate medications (PIMs) for older people (with or without dementia) has been reported previously in studies of residents in aged care facilities [10,11] and primary health care [3,12,13] contexts. PIMS may be defined as medications that pose potential risks that outweigh potential benefits [14]. People with dementia are vulnerable to PIMS because of increased sensitivity to centrally acting medications [14]. The prevalence of PIMs (or inappropriate drug use) has been reported to be 10 to 56% in the community overall, and higher in aged care facilities (36%) than in community dwelling people with dementia and cognitive impairment (28%) [14]. Other studies have reported this problem for hospitalized older people [2,4,8,15,16,17,18,19], however few studies have focused exclusively on people with dementia admitted to hospital [15,20]. The prevalence of PIMs for older inpatients has been reported to range from 53 to 90% and 30 to 97% respectively for admissions with and without cognitive impairment [15]. The problem of PIMs has specific risks for PWD because some PIMs may be used to manage behavioural problems and may exacerbate confusion for PWD. The use of PIMs for this purpose can worsen their cognitive function and increase their risk for falls, adverse drug events, admission to hospital and associated sequelae [14,21].

Many PIMs are known to contribute to anticholinergic cognitive burden, and this can increase the risk of dementia and other poor outcomes [22,23]. Examples of widely recognized scoring systems for anticholinergic burden and or PIMs include: the Anticholinergic Cognitive Burden Scale [24], the Scottish NHS mARS categories (http://www.polypharmacy.scot.nhs.uk/hot-topics/anticholinergics/), the European Union (7)-PIM list [21,25], the American Geriatric Society (AGS) Beers Criteria [26] and the STOPP (Screening Tool of Older Person’s Prescriptions) and START (Screening Tool to Alert doctors to Right Treatment) [27].

Some previous studies measuring PIMs in hospital admissions of older people [2,4,16,18] and dementia patients [20], have compared PIMs (using STOPP/START criteria or Beers criteria) at admission and discharge. The frequency of PIMs was reported to range from 39 to 77% at admission [2,4,18], and 20 to 71% at discharge [2,16,18] and in studies that reported PIMS at both time points, PIMs increased by 3 to 7% at discharge [2,18]. Only one study (*n* = 118) reporting PIMS (using Revised Beers Criteria (RBC)) for PWD admitted to a geropsychiatry hospital unit, found that PIMs decreased significantly between admission and discharge, and the mean number of Revised Beers Criteria medications per patient changed from 0.8 to 0.4 (*p* = 0.01) [20]. Admission to hospital presents a valuable opportunity to recognize PIMs for PWD and to review and modify their medications prior to discharge. In addition, when PWD are admitted to hospital, they frequently experience exacerbations of confusion and disorientation that are challenging for acute care staff to manage, and are often prescribed sedatives, antipsychotics or anticholinergics as well as having medication changes during hospitalization. This study aimed to compare PIMs at admission and at discharge for people with dementia or cognitive impairment admitted to hospital, to determine whether PIMS were identified during the provision of usual care during an acute admission and whether they changed prior to discharge. The overall aim of this study is to conduct a trial to evaluate a practical intervention to improve the care and support of people with dementia transitioning from acute care to the community and this paper reports medication data from the pre-intervention phase.

## 2. Materials and Methods

### 2.1. Study Objectives


(1)To compare polypharmacy and PIMs at admission and discharge, and by study site, and determine whether these changed during admission.(2)To compare anticholinergic burden at admission and discharge, and determine whether these changed during admission.(3)To compare polypharmacy and PIMs at admission for participants admitted from home with admissions from Residential Aged Care Facilities (RACF).


### 2.2. Study Design and Study Setting

This prospective quasi-experimental pre/post-controlled trial was conducted at two regional hospitals providing usual care in the Hunter New England Local Health District/New South Wales, Australia over one year (October 2017 to September 2018); and data reported in this paper are from phase 1 (pre-intervention phase). PWD or cognitive impairment having an unplanned admission to hospital were recruited to the study. Medications documented in patient’s records at admission and discharge were evaluated for these participants.

### 2.3. Participant Eligibility

Potential participant eligibility was determined as follows:-Admission reports were received by the researchers each day containing a list of all newly admitted inpatients over the age of 50 years who were admitted to general medicine, cardiology and general surgical admissions (inpatients).-Admissions were excluded if they had been previously recruited as participants in the study or if they were inpatient transfers from other hospitals.-Admissions were included if they were admitted via the emergency department from home, residential aged care facilities (RACF) or if they were acute admissions transferred from the emergency department of another Hunter New England Local Health District hospital.-Admissions with a history of dementia OR Mini Mental State Examination (MMSE) score <24/30 in the emergency department; or documented in Aged Care Service Emergency Team (ASET) nurse notes, or Aged Care Assessment Team (ACAT)/Geriatrician or Community Assistance Program (CAP) community notes; were determined to be eligible participants for the study.-Inpatient notes of all patients >75 years of age were checked for a history of dementia; and if it was not documented, they were screened for memory and confusion problems using the Six Item Screener [28] and the Confusion Assessment Method Instrument (CAMI) [29] instruments.

After eligibility checking was conducted using these instruments, patients who scored 4/6 or less from the Six Item Screener (for memory problems); AND/OR scores of 1, 2 & 3 or 1, 2 & 4 from the CAMI (for assessing confusion) were determined to be eligible participants for the study.

### 2.4. Recruitment of Patients to Participate in the Study

Recruitment and consent procedures for eligible admissions were conducted by study nurses employed for the study at both study sites. Because these patients had cognitive problems, we sought substitute consent from the person responsible for the patient. This required the use of the NSW Guardianship Act 1987 (Part 5: Substitute Consent: What the law says) [30] hierarchy of persons responsible to determine the appropriate person to approach, to seek consent for the patient to participate in the study.

**Confirming eligibility**: After the person responsible was identified, project staff contacted them and explained staff had determined the patient had confusion and memory problems and asked the person responsible *whether the patient had confusion and memory problems at home*. This question was asked to exclude potential participants who may have been suffering transient delirium rather that dementia or ongoing cognitive problems. If the person responsible confirmed the patient had confusion and/or memory problems at home, they were provided with information about the study and an invitation was extended for their relative to be a participant in the study. After they had time to consider the invitation, and if they were agreeable, they were invited to sign the consent form for the patient/their relative to be a participant in the study.

### 2.5. Study Sample

Sample size calculations were based on the primary aim (treatment effect for readmissions/re-presentation to ED) for the prospective quasi-experimental pre/post-controlled trial; and this examination of medication use within the pre-intervention phase is a secondary outcome, and as such, sample size calculations were not performed for this outcome. The final sample for the pre-intervention phase comprised 277 participants with dementia or cognitive impairment.

### 2.6. Data Collection

Data were collected at two time points for each participant: after admission as an inpatient (usually within 48 h) and prior to discharge (usually within 24 h). Data collected after admission included: demographic data, eligibility data, medications on admission; and data collected prior to discharge included medications at discharge. Other medication data included: PIMs (Based on Beers Criteria [26], and categorised for use for PWD), Modified Anticholinergic Burden Scale (mACB) (AUS) (Based on the Anticholinergic Cognitive Burden Scale [24] and the Scottish NHS mARS Scale (http://www.polypharmacy.scot.nhs.uk/hot-topics/anticholinergics/) and modified for medications approved and in current use in Australia) (Appendix A). Study data were collected and managed using REDCap electronic data capture tools hosted at the Hunter Medical Research Institute, Australia. REDCap (Research Electronic Data Capture) [31] is a secure, web-based application designed to support data capture for research studies, providing: (1) an intuitive interface for validated data entry; (2) audit trails for tracking data manipulation and export procedures; (3) automated export procedures for seamless data downloads to common statistical packages; and (4) procedures for importing data from external sources.


**Definitions:**
-Documentation of diagnosis of cognitive impairment or dementia was sourced from several medical record sources including history of dementia-related ICD-10 code (F00, F00.0, F00.1, F00.2, F00.9, F01, F01,1, F01.2, F01.3, F01.8, F01.9, F02.3, F02.4, F02.8, F03, F05.1), ASET notes, triage notes, CAP community notes and correspondence, Aged Care Assessment Team (ACAT)/geriatrician/community dementia notes, or current admission notes.-The number of medications, number of potentially inappropriate medications (PIMs), proton pump inhibitors (PPIs), and nonsteroidal anti-inflammatory drugs (NSAIDs); and total modified anticholinergic burden score (mACB score) were calculated for each patient. PIMs subcategories of psychotropics, and sedatives/hypnotics were examined separately. For each patient, the prescription of at least one PIM, psychotropics, and sedatives/hypnotics was recorded as Yes/No. Polypharmacy was categorised as taking ≥5 medications.


### 2.7. Data Analysis

Demographics and admission-related characteristics are presented as count (%), mean (standard deviation; SD), or median (quartile 1, quartile 3; Q1 and Q3). Age was compared between admission source (home vs. RACF) using independent t-test. Medication use (number of medications, number of PIMS + subcategories, mACB score) is presented as mean (SD) and median (min, max) by time (admission or discharge). Polypharmacy (Y/N) and the prescription of (at least 1) PIMs + subcategories, proton pump inhibitors (PPIs) and nonsteroidal anti-inflammatory drugs (NSAIDs), is presented as count (%). Differences between admission and discharge medications (number of medications, number of PIMS + psychotropics and sedatives/hypnotics, mACB score) were examined using linear mixed modelling. Assumptions for linear modelling (normality of residuals, homoscedasticity of residuals, outliers and influential data) were examined; if minor violations of normality of residuals were observed empirical standard errors were used. Time (discharge vs admission) was included in modelling as a fixed effect, and a random intercept for participant ID was included to account for paired measurements at the 2 time points. Confounding and effect modification of change over time for each medication measure was examined based on study site, and admission source. An interaction term for time*covariate was included in modelling (with main effects) to examine effect modification; if this was non-significant, it was removed to examine confounding. The proportions of polypharmacy (Y/N), PIMS (Y/N), psychotropics (Y/N), and sedatives/hypnotics (Y/N) were compared between admission and discharge using binary logistic mixed modelling, with modelling performed as described above. Statistical analyses were programmed using SAS v9.4 (SAS Institute, Cary, NC, USA). A priori, *p* < 0.05 (two-tailed) was used to indicate statistical significance.

### 2.8. Ethical Approval

This study was approved by the Hunter New England Health Human Research Ethics Committee (HREC) (17/06/21/4.08) and University of Newcastle (Australia) HREC (H-2017-0260). All participants had written consent provided by their carer or person responsible prior to their participation in the study.

## 3. Results

### 3.1. Demographic and Admission Characteristics

In Phase 1 (pre-intervention phase) for the SMS study, 2748 participants were screened, 365 (13.3%) were eligible, and of those, 277 (75.9%) consented to participate in the study. Demographics and admission characteristics are presented in Table 1.

There were similar numbers of male and female admissions, and 79% were admitted from home. The mean age was 84 and when age was examined by admission source, patients from RACF were significantly older than patients from home (87 vs. 84; *p* = 0.004).

### 3.2. Medication Use over Time

At admission, 3131 medications were recorded, which reduced to 2361 at discharge. Of these, at admission, 1091 (34.9%) were PIMs, and at discharge 838 (35.5%) were PIMs. Medication use (number of medications, polypharmacy, PIMs, and mACB score) was examined and compared at admission and discharge (Table 2).

The mean number of medications was 10 (range 1 to 27), the proportion of patients with polypharmacy was 92%, and the proportion prescribed at least one PIM was 91%. Overall, patients had (two) less medications and (one) less PIM, and a lower mACB score at discharge.

A significant decrease was seen in number of medications, PIMs, and mACB score between admission and discharge, but not in psychotropics or sedatives/hypnotics (Table 3).

A significant reduction in the odds of taking at least one PIMs medication, and at least one sedative/ hypnotic medication was seen from admission to discharge (Table 4). A reduction was seen in the odds of having polypharmacy, and being on at least one psychotropic medication from admission to discharge, however this was not found to be statistically significant.

PIMs prescription was examined by classification (Table 5).

The most frequently prescribed PIM classifications were: Other medications with postural hypotension as a side effect, PPI’s, anticoagulants and psychotropics. There was a reduction by half in anticoagulants and sedatives/hypnotics between admission and discharge. NSAIDs were examined by admission source; at admission 24/277 (8.7%) had been prescribed NSAIDs, and at discharge 8/277 (2.9%) had been prescribed NSAIDs; overall 32/554 (5.8%).

#### 3.2.1. Site and Medication Use over Time

The change in medication use over time was examined by site (Table 6 and Table 7).

The interaction term between change over time and site was non-significant for all linear and logistic mixed models indicating that the change over time (admission to discharge) was consistent for all medication use measures by site; site was therefore included in the model as a covariate to examine whether the change over time estimate was confounded by site. While number of medications, PIMs (number and taking at least 1), and mACB score were significantly higher at site 1 than at site 2, over time adjusted regression estimates did not change substantially when compared to crude regression estimates indicating no confounding due to site.

#### 3.2.2. Admission Source and Medication Use over Time

The change in medication use over time was examined by admission source (Table 8 and Table 9).

An interaction between change over time and admission sources was seen for number of medications; patients from RACF had a higher number of medications at admission, and a larger reduction in number of medications over time. Patients from RACF still had higher number of medications than patients from home at discharge. No interaction was seen between changes over time and admission source for all other medication use measures; the interaction term was removed, and admission source was included in the models as a covariate. Number of PIMs, mACB score, and psychotropic medication (at least 1) was significantly higher in participants admitted from RACF than from home, however over time, adjusted regression estimates did not change substantially when compared to crude regression estimates indicating no confounding due to admission source.

## 4. Discussion

Participant characteristics indicated that these patients were elderly and those from RACF were significantly older than those from home. Most admissions were from home (79%) and this suggests that many of them may have been in the earlier stages of dementia [32]. Dementia or cognitive impairment was routinely documented for 75%, and memory or confusion problems were only documented for 28% of these admissions. These results indicate that the cognitive status of these admissions may often not be detected during an unplanned admission, and this makes PWD more vulnerable to medication related problems.

The average number of medications at admission (11.5) was significantly reduced at discharge (9.3), *p* < 0.0001. Admissions from RACF had a significantly higher number of medications at admission and discharge compared with admissions from home (Table 8). In contrast, these results might be compared with a previous study of 1187 inpatients (>70 years, mean age 81 years, 30% with cognitive impairment) that reported a significant increase in mean number of medications per day between admission and discharge (7.1–7.6), and a decrease in the number of medications with higher prevalence of severe cognitive impairment [1]. Another study also reported an increase in the number of medications at discharge (9.1–10.1, *p* < 0.001) for 200 inpatients from an acute geriatric unit [2].

When polypharmacy was examined in this study, 94% of admissions were taking >5 medications. Although polypharmacy reduced at discharge (90%), there was not a significant difference. While polypharmacy for PWD might be expected to be higher, the study by Hubbard et al., reported polypharmacy for 69% of participants with cognitive impairment on admission [1]. In the study of 200 admissions to the acute geriatric unit, polypharmacy reduced from 87 to 82% [2]. The results from this study suggest that while some medications were reduced during admission to hospital, there is potential for further reductions [2], particularly where potentially inappropriate prescribing is an issue.

The number of admissions prescribed at least one PIM was 96% at admission and this reduced to 87% at discharge, and the mean number of PIMS reduced significantly from 4 to 3.3 (*p* < 0.0001). A significant reduction in the odds of taking at least one PIM was determined between admission and discharge. This result is higher than a previous study of patients in an acute geriatric unit (*n* = 200) that reported an increase in PIMs (68.5–71.5%) using Beers and STOPP/START criteria [2]; a study of 300 patients in an acute medical geriatric division that reported a significant increase in PIMs (39.3–46%, *p* = 0.009) using STOPP/START criteria [18]; and a study of 1380 inpatients (>65 years) that reported PIMs prevalence of 20.1 to 23.5% using different versions of Beers criteria [16]. However, none of these previous studies of inpatients were conducted specifically on admissions of PWD, and the high prevalence reported for this study indicates increased risk associated with PIMs for these admissions [14]. The results are comparable to the prevalence of PIMs in RACF where PIMs prevalence was reported as 81.4% for 533 participants and 81.3% for participants (461) with cognitive impairment and dementia [11]; and higher than the prevalence of PIMs in primary care for 38,229 older people (45.3 to 51%) and the odds of PIMs after admission was higher OR 1.72 (1.63 to 1.84) [33]. A European study of PWD in long term institutional care and home care, reported that 60% of 2004 PWD had at least one PIM using the EU(7)-PIM list [25].

Behavioural and psychological symptoms of dementia (BPSD) are frequently treated with psychotropics including antipsychotics however, it has been reported that only 20% derive clinical benefit from them [34,35]. Recent consensus recommendations for managing BPSD include non-pharmacological strategies as a first-line response and antipsychotics only for a limited period if symptoms are not responsive to other strategies [35,36,37]. Sedatives/hypnotics such as benzodiazepines are also not generally recommended for managing BPSD, and if used they should be monitored [36]. In this study, psychotropics and sedatives/hypnotics were also compared, and a significant reduction in the odds of taking at least one sedative/hypnotic was found between admission and discharge. This is a positive result compared to a study of PIMS in RACF for PWD and cognitive impairment that reported 36% were prescribed benzodiazepines [11]. No difference was found for psychotropics, however their prevalence was 38% suggesting they were overprescribed for this study population and that they should have been reduced at discharge. The most frequently prescribed medication classifications indicate that many of these admissions had cardiovascular disease. The reason for the high number of PPIs prescribed (46%) is unknown and does not appear to be related to prescription of NSAIDs, and is comparable to prevalence of PPIs reported in RACF [11]. Anticoagulants included venous thromboembolism (VTE) prophylaxis and were reduced at discharge. Psychotropics (38%) included antipsychotics and antidepressants and some of these may have been used to manage BPSD during hospitalization, and this result is comparable to a study of PIMs in RACF (*n* = 461) that reported a rate of 34.1% for antipsychotics and 4.8% for antidepressants for PWD [11]; and lower than the rate of psychotropics reported for 1005 PWD in nursing home residents (76.2%) [38]. There is scope for pharmacists to provide information about alternate medications for PWD to minimize PIMs for these patients and risperidone is the only newer antipsychotic with a Therapeutic Goods Administration-approved indication for use in behavioural disturbance (in moderate to severe Alzheimer’s dementia) and is subsidized under the Pharmaceutical Benefits Scheme [35,39].

The mean mACB score also reduced significantly between admission (2.7) and discharge (2.3), *p* < 0.0001. Increased anticholinergic burden is reported to increase the risk of adverse events and dementia or cognitive impairment in older people [22,24,40]. In this study, the mACB score reduction was consistent with the reduction in PIMs (which include anticholinergic medications). Overall the mACB score was 3 (range 0 to 15), and this is a clinically significant result for PWD because a cumulative score of 3 or more is reported to indicate a potential harmful anticholinergic burden [41].

Analysis of polypharmacy and PIMs by site indicated that there was no effect modification or confounding of the over time effect due to study site. Analysis of number of medications by admission source found that these were higher at admission, and that there was a larger reduction in medications over time, for admissions from RACF. In addition, the number of PIMS, mACB score and psychotropic medications was significantly higher in admissions from RACF; however there was no confounding of the over time effect due to admission source. These results are consistent with the view that RACF residents are older, have more comorbidities and may have later stages of dementia. However, PWD in primary care have been reported to have a higher burden of disease and polypharmacy overall [5] and PIMs [14].

People with dementia are particularly vulnerable to poor outcomes associated with medications. The results of this study indicate that polypharmacy and PIMS prevalence for inpatients with dementia are issues that require more targeted attention to minimize the risks for PWD. One strategy that can be used to address this problem is pharmacist medication reviews during hospitalization, and in RACF and primary care settings [12]. This study has identified that an admission to hospital presents an opportunity to address polypharmacy and PIMs for PWD, however their cognitive status may not always be identified at admission, and hospital pharmacists may not routinely have an opportunity to conduct a medication review if they are not notified about these high-risk patients. The significant reductions in polypharmacy, PIMs and mACB scores in this study are encouraging, particularly when compared with previous studies that reported increases in these at discharge. However, PWD should be identified as high risk at admission, and should be provided with medication reviews to specifically target these issues and improve medication safety for PWD at discharge.

## 5. Conclusions

This study indicates that unplanned admissions of people with dementia or cognitive impairment to hospital present a good opportunity to intervene on medication issues, particularly for those from RACF who are on substantially more medications. Current general medical practice already includes some de-prescribing. Whether this can be enhanced will be the subject of the intervention phase of this study, which will include specialist pharmacist input.

## Figures and Tables

**Table 1 healthcare-07-00008-t001:** Demographics and admission characteristics.

Characteristic	Class/Statistic	Total (*n* = 277)
Study Site	Site 1	160 (58%)
	Site 2	117 (42%)
Age at admission	mean (SD)	84 (7)
Sex	Male	138 (50%)
	Female	139 (50%)
Aboriginal status	Aboriginal	3 (1.1%)
	Neither	273 (99%)
Speciality	General medicine	243 (88%)
	Cardiology	27 (9.7%)
	General surgery	7 (2.5%)
Admitted from	Home	220 (79%)
	RACF	57 (21%)
Dementia/cognitive impairment documented	No	70 (25%)
	Yes	207 (75%)
Memory/confusion documented	No	200 (72%)
	Yes	77 (28%)

**Table 2 healthcare-07-00008-t002:** Medication use at admission and discharge.

Medication Outcome	Measure	Admission (*n* = 277)	Discharge (*n* = 277)	Total (*n* = 554)
Number of medications	mean (SD)	11 (5)	9 (4)	10 (4)
	median (min, max)	11 (1, 27)	9 (1, 22)	10 (1, 27)
Polypharmacy (≥5 meds)	No	17 (6.2%)	25 (9.8%)	42 (8.0%)
	Yes	256 (94%)	229 (90%)	485 (92%)
PIMs	mean (SD)	4 (2)	3 (2)	4 (2)
	median (min, max)	4 (0, 11)	3 (0, 9)	3 (0, 11)
PIMs (at least one)	No	11 (4.0%)	37 (13%)	48 (8.7%)
	Yes	266 (96%)	240 (87%)	506 (91%)
PIMs (psychotropics)	mean (SD)	1 (1)	1 (0)	1 (1)
	median (min, max)	1 (1, 4)	1 (1, 3)	1 (1, 4)
PIMs (psychotropics: at least 1)	No	170 (61%)	176 (64%)	346 (62%)
	Yes	107 (39%)	101 (36%)	208 (38%)
PIMs (sedatives/hypnotics)	mean (SD)	1 (0)	1 (0)	1 (0)
	median (min, max)	1 (1, 2)	1 (1, 2)	1 (1, 2)
PIMs (sedatives/hypnotics: at least 1)	No	240 (87%)	259 (94%)	499 (90%)
	Yes	37 (13%)	18 (6.5%)	55 (9.9%)
mACB score	mean (SD)	3 (2)	2 (2)	3 (2)
	median (min, max)	2 (0, 15)	2 (0, 11)	2 (0, 15)

**Table 3 healthcare-07-00008-t003:** Change over time in medication number/score; linear mixed modelling.

Medication Outcome	Estimated Mean (Std. Error)		Regression Estimates Difference (95% CI)	*p*-Value
	Admission	Discharge		
Number of medications	11.5 (0.3)	9.3 (0.3)	−2.2 (−2.6, −1.8)	<0.0001
PIMS	4.0 (0.1)	3.3 (0.1)	−0.7 (−0.9, −0.5)	<0.0001
Psychotropics *	1.3 (0.1)	1.2 (0.0)	−0.1 (−0.2, 0.0)	0.2853
Sedatives/hypnotics *	1.1 (0.0)	1.1 (0.1)	0.1 (−0.1, 0.2)	0.4910
mACB score *	2.7 (0.1)	2.3 (0.1)	−0.4 (−0.6, −0.2)	<0.0001

* Empirical standard errors applied to linear mixed modelling.

**Table 4 healthcare-07-00008-t004:** Change over time in medication use (Y/N).

Medication Outcome	Regression Estimates Odds Ratio (95%CI)	*p*-Value
Polypharmacy	0.59 (0.30, 1.16)	0.1240
PIMs YN	0.25 (0.12, 0.51)	0.0002
Psychotropics YN	0.88 (0.59, 1.32)	0.5380
Sedatives/hypnotics YN	0.43 (0.24, 0.80)	0.0072

**Table 5 healthcare-07-00008-t005:** Prescription of (at least 1) PIM, by classification.

PIMs Classification	Admission (*n* = 277)	Discharge (*n* = 277)	Total (*n* = 554)
Anticholinergics	45 (16%)	46 (17%)	91 (16%)
Anticoagulants	171 (62%)	81 (29%)	252 (45%)
Anticonvulsants	45 (16%)	43 (16%)	88 (16%)
Antiemetics	29 (10%)	10 (3.6%)	39 (7.0%)
Long term opioid analgesics	42 (15%)	40 (14%)	82 (15%)
Other parkinson’s	17 (6.1%)	18 (6.5%)	35 (6.3%)
Other postural HTN	210 (76%)	191 (69%)	401 (72%)
Psychotropics	107 (39%)	101 (36%)	208 (38%)
Sedating antihistamines	6 (2.2%)	4 (1.4%)	10 (1.8%)
Sedatives/hypnotics	37 (13%)	18 (6.5%)	55 (9.9%)
Sliding scale insulin	7 (2.5%)	0 (0%)	7 (1.3%)
Sulfonylureas	18 (6.5%)	16 (5.8%)	34 (6.1%)
PPI	130 (47%)	126 (45%)	256 (46%)

Anticoagulants included heparins, direct thrombin inhibitors, factor Xa inhibitors, low molecular weight heparin and others. Other medications with postural hypotension as a side effect included diuretics (loop diuretics, thiazide and related diuretics); antihypertensives (alpha-adrenergic agonists, angiotensin converting enzyme inhibitors, calcium channel blockers, alpha-blockers, angiotensin II antagonists, vasodilators, beta-blockers and others); nitrates; phosphodiesterase inhibitors (antiplatelet); phosphodiesterase 5 inhibitors; and others. Psychotropics included antipsychotics and antidepressants. Sedatives/hypnotics included benzodiazepines and imidazopyridine hypnotics.

**Table 6 healthcare-07-00008-t006:** Medication use over time by site; linear mixed modelling.

Medication Outcome	Comparison	Estimated Mean (Std. Error)		Regression Estimates Difference (95% CI)	*p*-Value
		Admission	Discharge		
Number of medications	Over time	11.6 (0.3)	9.4 (0.3)	−2.2 (−2.6, −1.8)	<0.0001
	Site 2 (vs. Site 1)			1.3 (0.3, 2.2)	0.0074
PIMS	Over time	4.1 (0.1)	3.3 (0.1)	−0.7 (−0.9, −0.5)	<0.0001
	Site 2 (vs. Site 1)			0.9 (0.4, 1.3)	0.0002
Psychotropics *	Over time	1.3 (0.1)	1.2 (0.0)	−0.1 (−0.2, 0.0)	0.2758
	Site 2 (vs. Site 1)			−0.1 (−0.3, 0.1)	0.3165
Sedatives/hypnotics *	Over time	1.1 (0.0)	1.1 (0.1)	0.1 (−0.1, 0.2)	0.4773
	Site 2 (vs. Site 1)			0.0 (−0.2, 0.2)	0.9614
mACB score *	Over time	2.7 (0.1)	2.4 (0.1)	−0.4 (−0.6, −0.2)	<0.0001
	Site 2 (vs. Site 1)			0.5 (0.0, 1.0)	0.0353

* Empirical standard errors applied to linear mixed modelling.

**Table 7 healthcare-07-00008-t007:** Medication use over time by site; binary logistic mixed modelling.

Medication Outcome	Comparison	Regression Estimates Odds Ratio (95%CI)	*p*-Value
Polypharmacy	Over time	0.59 (0.30, 1.15)	0.1235
	Site 2 (vs. Site 1)	1.50 (0.69, 3.27)	0.3016
PIMs YN	Over time	0.24 (0.12, 0.50)	0.0001
	Site 2 (vs. Site 1)	2.31 (1.06, 5.04)	0.0345
Psychotropics YN	Over time	0.88 (0.59, 1.32)	0.5366
	Site 2 (vs Site 1)	1.43 (0.80, 2.55)	0.2209
Sedatives/hypnotics YN	Over time	0.43 (0.24, 0.79)	0.0071
	Site 2 (vs Site 1)	1.36 (0.71, 2.60)	0.3569

**Table 8 healthcare-07-00008-t008:** Medication use over time by admission source; linear mixed modelling.

Medication Outcome	Comparison	Estimated Mean (Std Error)		Regression Estimates Difference (95%CI)	*p*-Value
		Admission	Discharge		
Number of medications	Admission source * time				0.0005
	Home-over time	10.8 (10.2, 11.3)	9.0 (8.4, 9.5)	−1.8 (−2.3, −1.4)	<0.0001
	RACF-over time	14.2 (13.1, 15.3)	10.5 (9.4, 11.6)	−3.7 (−4.6, −2.7)	<0.0001
	Admission source-at admission			3.4 (2.2, 4.6)	<0.0001
	Admission source-at discharge			1.6 (0.3, 2.8)	0.0149
PIMs	Over time	4.2 (0.2)	3.5 (0.2)	−0.7 (−0.9, −0.5)	<0.0001
	RACF (vs. Home)			0.8 (0.3, 1.4)	0.0035
Psychotropics *	Over time	1.3 (0.1)	1.3 (0.1)	−0.1 (−0.2, 0.0)	0.2883
	RACF (vs. Home)			0.1 (−0.1, 0.4)	0.1829
Sedatives/hypnotics*	Over time	1.0 (0.0)	1.1 (0.1)	0.1 (−0.1, 0.2)	0.4215
	RACF (vs. Home)			−0.0 (−0.2, 0.1)	0.5939
mACB score*	Over time	2.9 (0.2)	2.5 (0.2)	−0.4 (−0.6, −0.2)	<0.0001
	RACF (vs. Home)			0.7 (0.0, 1.3)	0.0493

* Empirical standard errors applied to linear mixed modelling.

**Table 9 healthcare-07-00008-t009:** Medication use over time by admission source; binary logistic mixed modelling.

Medication Outcome	Comparison	Regression Estimates Odds Ratio (95%CI)	*p*-Value
Polypharmacy	Over time	0.59 (0.30, 1.15)	0.1225
	RACF (vs. Home)	1.55 (0.56, 4.29)	0.3975
PIMs YN	Over time	0.25 (0.12, 0.51)	0.0002
	RACF (vs. Home)	1.14 (0.47, 2.80)	0.7699
Psychotropics YN	Over time	0.88 (0.58, 1.32)	0.5352
	RACF (vs. Home)	2.11 (1.05, 4.24)	0.0357
Sedatives/hypnotics YN	Over time	0.43 (0.23, 0.79)	0.0069
	RACF (vs. Home)	1.84 (0.88, 3.83)	0.1032

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
