# Peer review of "Comparison of Potentially Inappropriate Medications for People with Dementia at Admission and Discharge during An Unplanned Admission to Hospital: Results from the SMS Dementia Study†"

_healthcare, 2019, doi:10.3390/healthcare7010008_

Round 1

Reviewer 1 Report

Thank you for the opportunity to review this manuscript on an important topic. I find the article of interest but I recommend some minor changes before it can be published, particularly in clarity in the introduction and methods. Please see below for specific comments. 

Introduction:

The first sentence "The issue of poly pharmacy for older people  has been the focus on many studies because of the potential for patients to not understand the indications for their medications...." is unclear - I think you mean the studies have shown the potential for patient to not understand the indications for their medications etc- please revise. 

There are some long sentences in the introduction which could be broken up to aid the reader e.g. "The problem of PIMS has specific risks for PWD because some PIMs may be used to manage behavioural problems and may exacerbate confusion....."

Please specific which scoring system you refer to e.g. scoring for anticholinergic risk? in this sentence "Examples of widely recognised scoring systems include...." and what these scoring systems are used for? 

Methods: In the participant eligibility section I am unclear about whether patients who were admitted with a history of dementia or a MMSE score <24 were automatically eligible for the study or if they were also screened using the Six Item Screener or CAMI? 

In the. methods section, please include the reference or appropriate attribution for the medication scoring systems used.  

Please spell out all abbreviations at first use e.g ASET notes, CAP community notes, Y/N. I don't think you need the separate 'definitions' section. This could be incorporated into the main text of the methods. under data collection or data analysis. 

Results: I note the difference in age between residential care and home admissions - were there are other differences? I assume not but would be good to state this if it is the case. For table 2, it would be useful to have some basic significance testing of any difference between admission and discharge. 

Author Response

Thank you for the opportunity to review this manuscript on an important topic. I find the article of interest but I recommend some minor changes before it can be published, particularly in clarity in the introduction and methods. Please see below for specific comments.

Response: Thank you very much.

Introduction:

The first sentence "The issue of poly pharmacy for older people  has been the focus on many studies because of the potential for patients to not understand the indications for their medications...." is unclear - I think you mean the studies have shown the potential for patient to not understand the indications for their medications etc- please revise. 

Response: This sentence has not been modified. It describes the reasons for conducting studies about polypharmacy for older people rather than the results of these studies.

There are some long sentences in the introduction which could be broken up to aid the reader e.g. "The problem of PIMS has specific risks for PWD because some PIMs may be used to manage behavioural problems and may exacerbate confusion....."

Response: The introduction has been checked and this long sentence has been amended.

Please specific which scoring system you refer to e.g. scoring for anticholinergic risk? in this sentence "Examples of widely recognised scoring systems include...." and what these scoring systems are used for? 

Response: This has been specified as requested.

Methods: In the participant eligibility section I am unclear about whether patients who were admitted with a history of dementia or a MMSE score <24 were automatically eligible for the study or if they were also screened using the Six Item Screener or CAMI? 

Response: Patients with a history of dementia or a MMSE score <24 were automatically eligible for the study. If this information was not documented, screening was undertaken using the Six Item Screener and CAMI. (Please see the coloured font section in 2.3 where this is described).

In the methods section, please include the reference or appropriate attribution for the medication scoring systems used.  

Response: References have been included for the medication scoring systems as requested.

Please spell out all abbreviations at first use e.g ASET notes, CAP community notes, Y/N.
I don't think you need the separate 'definitions' section. This could be incorporated into the main text of the methods. under data collection or data analysis. 

Response: The paper has been checked for abbreviations and where they are used first, they have been spelled out as requested. The definitions are included under Data Collection.

Results: I note the difference in age between residential care and home admissions - were there are other differences? I assume not but would be good to state this if it is the case.

Response: There were also differences between residential care and home admissions in the number of medications (higher number of medications for admissions from RACF) however, after analysis it was determined there was no confounding due to the admission source. This is reported in the paragraph immediately following table 9.
These results are consistent with the view that RACF residents are older, have more comorbidities and may have later stages of dementia.

For table 2, it would be useful to have some basic significance testing of any difference between admission and discharge. 

Response: The significance testing for items in table 2 has been presented in table 3. Data items in table 3 that are significant include: Number of medications, PIMS, and mACB score.

Reviewer 2 Report

This is a well-written study, but some revision is needed.

Abstract

Data collection tool, setting and sample should be described.

Introduction

Please provide some context-based details of the motivation behind conducting this study.

Methods

Please reorganize this section using the elements and headings of CONSRT. Also, please add the CONSORT flow diagram to the text.

How did you practically recruit the samples?

How about random selection?

Other parts are okay.

Author Response

This is a well-written study, but some revision is needed.

Response: Thank you very much.

Abstract

Data collection tool, setting and sample should be described.

Response: The abstract has been modified to include these items.

Introduction

Please provide some context-based details of the motivation behind conducting this study.

Response: A sentence has been included to provide this information at the end of the introduction.

Methods

Please reorganize this section using the elements and headings of CONSRT. Also, please add the CONSORT flow diagram to the text.

Response: This study was a quasi-experimental pre/post-controlled trial. It was not a randomised controlled trial. There was no randomised allocation of participants in this study. There was a control site and a study site. This is reported in the manuscript in section 2.2.

How did you practically recruit the samples?

Response: Study nurses were employed to undertake participant recruitment. They screened daily admission lists to determine potentially eligible participants (patients). Participant eligibility is described in section 2.3. Then they determined the person responsible for these patients and contacted them to confirm whether the patient had confusion and memory problems at home. This question was used to exclude potential participants who may have been suffering transient delirium rather than dementia or ongoing cognitive problems. If the person responsible confirmed the patient had confusion/ and or memory problems at home, the study nurse provided them with information about the study and extended an invitation for their relative to be a participant in the study. This process is reported in section 2.4.

How about random selection?

Response: Random selection of participants was not feasible for this study due to the complex organisational processes associated with admission and management of people with dementia and memory and confusion problems, and the fact that they were not admitted to a dedicated dementia ward or unit at the study sites. Consequently, the study was designed as a prospective quasi-experimental pre/post-controlled trial, with a control site and an intervention site.

Other parts are okay.

Reviewer 3 Report

Authors investigated the polypharmacy and issues of potentially inappropriate medications that can happen when people with dementia are admitted to hospitals in unplanned manner. Further, anticholinergic burden was analyzed in this context. Patients with dementia possess cognitive impairments rendering them susceptible problems that can cause immediate urgent hospital care. Unplanned admit to hospitals create possibility of inappropriate medications such as anticholinergic burden. Authors investigated a comparative study of potentially inappropriate medications for Patients with dementia to infer if these changed during hospitalization. Medications discharge from Patients with dementia was evaluated during phase one of a quasi-experimental pre/post-controlled trial conducted hospitals in NSW, Australia. Since 277 participants were included in this study, we can have confidence on results. Results and analysis show that the cognitive status of Patients with dementia is not always detected during an unplanned hospital admit. This in fact increases the vulnerability to medication problems that can subsequently lead to poor outcomes.

Polypharmacy and Potentially Inappropriate Medications for Patients with dementia was found to be high at admission and later was significantly reduced at discharge. Authors concluded that patients with Dementia should undergo a routinely check for high risk at admission. This regime can potentially further reduce the polypharmacy and inappropriate medications such as psychotropic medications at discharge. The present paper is well articulated and contains sufficient analysis for acceptance at MDPI healthcare.

Author Response

Authors investigated the polypharmacy and issues of potentially inappropriate medications that can happen when people with dementia are admitted to hospitals in unplanned manner. Further, anticholinergic burden was analyzed in this context. Patients with dementia possess cognitive impairments rendering them susceptible problems that can cause immediate urgent hospital care. Unplanned admit to hospitals create possibility of inappropriate medications such as anticholinergic burden. Authors investigated a comparative study of potentially inappropriate medications for Patients with dementia to infer if these changed during hospitalization. Medications discharge from Patients with dementia was evaluated during phase one of a quasi-experimental pre/post-controlled trial conducted hospitals in NSW, Australia. Since 277 participants were included in this study, we can have confidence on results. Results and analysis show that the cognitive status of Patients with dementia is not always detected during an unplanned hospital admit. This in fact increases the vulnerability to medication problems that can subsequently lead to poor outcomes.

Polypharmacy and Potentially Inappropriate Medications for Patients with dementia was found to be high at admission and later was significantly reduced at discharge. Authors concluded that patients with Dementia should undergo a routinely check for high risk at admission. This regime can potentially further reduce the polypharmacy and inappropriate medications such as psychotropic medications at discharge. The present paper is well articulated and contains sufficient analysis for acceptance at MDPI healthcare.

Response: Thank you for these positive comments.